 

# Deciphering strain differences in codY regulation of *Clostridioides difficile*s sporulation

Marcos P. Monteiro,[1] Adrianne N. Edwards,[2] Michael A. DiCandia,[2] Shonna M. McBride[1]

**ABSTRACT**  *Clostridioides difficile* is an anaerobic, spore-forming pathogen that causes diarrhea, colitis, and even death. *C. difficile* grows and replicates in the intestine as a vegetative bacillus, but must transition into a dormant spore to survive and transmit in the environment. The transformation into a spore is a complex developmental process that is regulated in response to conditions within the host, most notably nutrient limitation. Nutrient availability is sensed by *C. difficile* through transcriptional regulators, such as CodY. CodY is a global nutritional gene regulator that controls gene expression in response to branched-chain amino acids (BCAAs) and guanosine-triphosphate (GTP). It was previously observed that CodY represses *C. difficile* sporulation, but the impact of CodY on sporulation has differed considerably by strain. Here, we investigated the effects of CodY on gene expression during sporulation in the two common research strains 630Δ*erm* (ribotype 012) and UK1 (ribotype 027). We confirmed that CodY suppressed premature spore formation in both strains through time-elapsed sporulation assays with *codY* mutants. Through transcriptional analyses of *codY* mutant sporulation, we defined the similarities and differences in the CodY-dependent gene expression between strains. We also identified differences in putative CodY sites within the 630 and UK1 genomes that may influence CodY regulation. Finally, we performed CRISPRi knockdowns to examine the effects of selected CodY-regulated genes, demonstrating the impact of multiple CodY-dependent factors on sporulation.

**IMPORTANCE**  *Clostridioides difficile* spore formation is crucial for transmission and survival of the bacterium. Spore formation is triggered by the availability of crucial nutrients, which CodY and other regulators sense. However, the mechanism by which CodY represses sporulation in *C. difficile* is poorly understood. In this study, we identified several CodY-regulated factors that could play a role in sporulation both in 630Δ*erm* and UK1 strains. Our results show that many factors under the regulation of CodY can impact sporulation.

**KEYWORDS**  *Clostridium difficile*, nutrient availability, CodY, sporulation

Address correspondence to Shonna M. McBride, shonna.mcbride@emory.edu.

The authors declare no conflict of interest.

See the funding table on p. 17.

Clostridioides difficile is an anaerobic and spore-forming nosocomial pathogen that causes severe diarrhea, colitis, and even death (1–3). Transmission of *C. difficile* is only possible through spores, which survive environmental threats, such as atmospheric oxygen and disinfectants (4). After a host ingests *C. difficile* spores, they transit through the gastrointestinal tract, reaching the intestines, where they sense bile salts and germinate into vegetative cells (5–7). *C. difficile* vegetative cells colonize the host colon, where nutrient availability is limited, leading to toxin production and spore formation (8–14). Nutrient availability is fundamental for determining whether *C. difficile* grows as a vegetative cell or becomes a spore. Under nutrient-limited conditions, *C. difficile* responds by increasing the expression of factors for nutrient acquisition and biosyn-

thesis of necessary metabolites; when these mechanisms fail to provide for sustained vegetative growth, spore formation is initiated (15–17).

To sense and control metabolism, *C. difficile* encodes nutritional regulators, such as the global nutrient transcriptional regulator, CodY (11, 13, 14, 18). CodY was first identified in *Bacillus subtilis* and is present in many gram-positive bacteria with low G-C genomes (19–26). In a nutrient-rich environment, *C. difficile* senses branched-chain amino acids (BCAAs) and guanosine triphosphate (GTP) through their interactions with CodY (11, 13, 14, 27, 28). CodY undergoes a conformational change when it binds to BCAAs and GTP, which increases its binding affinity to specific CodY-DNA binding sites, leading to the differential regulation of hundreds of genes (11, 13, 14, 29). When the intracellular concentrations of BCAAs and GTP decrease, the binding affinity of CodY to DNA is altered, changing gene expression to adapt to nutrient scarcity, including the derepression of toxin production and the initiation of sporulation (11, 13, 14, 29–31). While the regulation of specific metabolic genes and toxins by CodY is well-documented, the mechanisms by which CodY affects *C. difficile* sporulation are less clear (29, 32). CodY has varied effects on sporulation in strains 630 (ribotype 012) and UK1 (ribotype 027), as evidenced by a modest increase in sporulation in a 630 *codY* mutant and robust hypersporulation in a UK1 *codY* mutant (13, 29). The CodY proteins encoded by these strains are identical and similarly expressed, leading us to ask how CodY differentially regulates sporulation outcomes in these strains.

In this study, we examined CodY-dependent gene regulation in the 630 and UK1 backgrounds to identify strain-specific differences in sporulation outcomes. Through transcriptional analysis and mapping of CodY-binding sites, we identified CodY-regulated factors that are differentially expressed in 630Δ*erm* and UK1 and contain a CodY-binding site in at least one strain. In addition, we demonstrated that transcriptional repression of several direct CodY-regulated factors in UK1 or UK1 *codY* impacts sporulation. These results illustrate how CodY regulation differs between the 630Δ*erm* and UK1 strains and demonstrate that many CodY-regulated factors can impact sporulation.

## RESULTS

### The impact of CodY regulation on sporulation is strain-dependent

In a previous work, we demonstrated that CodY represses the initiation of sporulation and that CodY regulation of sporulation varies by strain (13). In the commonly studied strain 630Δ*erm* (a 630 derivative), CodY was found to modestly repress sporulation, resulting in a twofold increase in sporulation frequency for the *codY* mutant in sporulation broth cultures. In contrast, the epidemic 027 isolate, the UK1 *codY* mutant, demonstrated more than 1,000-fold greater sporulation frequency than the parent strain. To better understand how CodY regulates sporulation dissimilarly in 630Δ*erm* and UK1, we evaluated sporulation in these strains over time on sporulation agar, which induces more robust sporulation than liquid medium (33, 34). Strains UK1, 630Δ*erm*, and their respective *codY* mutants were grown on 70:30 sporulation agar, and the formation of ethanol-resistant spores was assessed after 6 (logarithmic phase), 12 (stationary phase), and 24 h of growth to compare the dynamics of spore production. As shown in Fig. 1, at log phase, the 630Δ*erm codY* mutant sporulated ~43-fold more than its parent strain ($1.0E{-}3 \pm 4.3E{-}4$ vs $2.6E{-}5 \pm 3.3E{-}5\%$). In comparison, at log phase, the UK1 *codY* mutant sporulated ~3,150-fold more than its parent strain, UK1. These results support the prior evidence that CodY represses premature sporulation initiation and that CodY repression of sporulation in UK1 is more robust than in 630Δ*erm* (13). By stationary phase (12 h), the 630 *codY* mutant sporulated ~28-fold less than the parent strain ($0.12 \pm 0.07$ vs $3.37 \pm 1.02\%$), while after 24 h of growth, the 630 *codY* mutant and parent displayed similar sporulation frequencies (Fig. 1). These results suggest that CodY suppresses early initiation of sporulation in 630 and that this strain requires CodY to control the timing of sporulation. In contrast, at stationary phase, the UK1 *codY* mutant sporulation frequency was ~2,000-fold higher than its parent strain ($45.4 \pm 16.9$ vs $0.02 \pm 0.01\%$) and continued at greater frequency than the parent at 24 h ($67.9 \pm 4.9$ vs $0.33 \pm 0.05$). Thus, the UK1

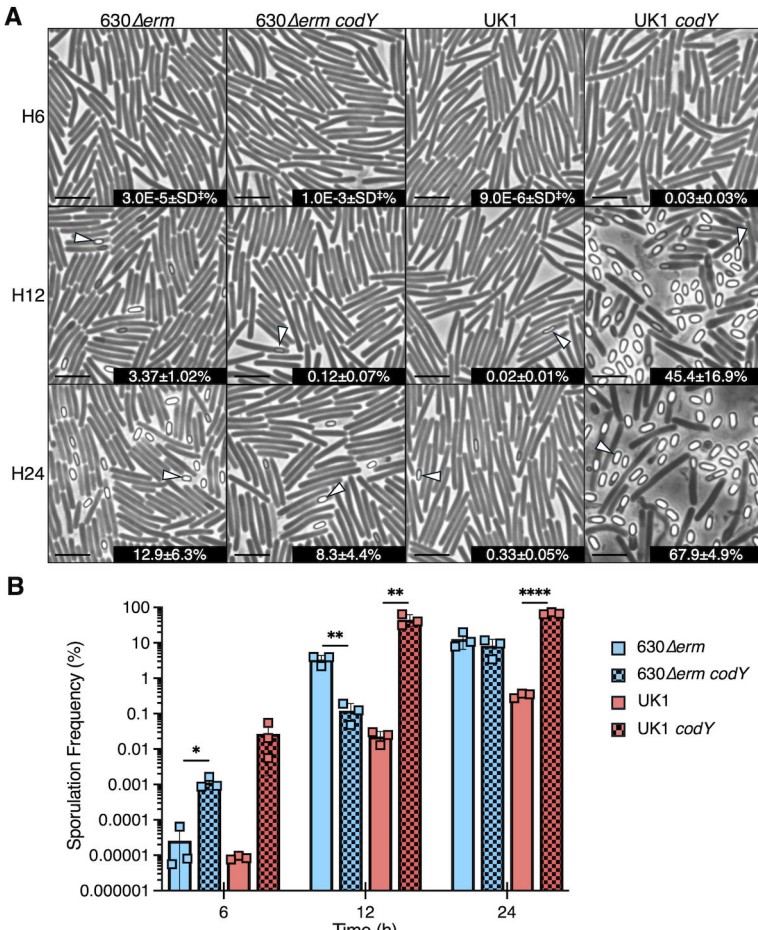

**FIG 1** CodY repression on sporulation is strain-dependent. (A) Phase-contrast micrographs of strains 630Δerm, 630Δerm codY (MC364), UK1, and UK1 codY (LB-CD16) grown on sporulation agar for 6, 12, or 24 h. White arrowheads indicate bright spores. Scale bar = 5 µm. ‡SD: standard deviation <0.0001. (B) Ethanol-resistant spore formation for the cultures above. The means and individual values for three biological replicates are shown. Data were analyzed using unpaired Student's t-tests comparing the mutants to their respective parent strain. *P < 0.05, ** P < 0.01, and **** P < 0.0001.

strain CodY represses sporulation at all growth stages. These data suggest there are differences in CodY-dependent gene regulation in U K1 and 630 that result in dissimilar sporulation outcomes.

## Identifying strain-specific differences in CodY regulation

To understand how CodY regulates sporulation differently in the UK1 and 630 backgrounds, we examined gene expression during growth on sporulation agar in these strains and their *codY* mutants. Since CodY activity is controlled by the availability of BCAA and GTP, we investigated expression at log phase when nutrients are most abundant, and CodY repression is greatest (11, 13, 14, 19, 27–29, 35, 36). Following 6 h of growth on 70:30 agar, samples were processed for RNA-seq analysis to assess the ratio of gene expression in the *codY* mutants relative to their respective parent strain (*codY*/WT) (Tables S1 and S2). Transcription was extensively altered in the *codY* mutants of both strains, resulting in 867 genes differentially expressed more than threefold in the UK1 *codY* mutant and 449 genes in the 630 *codY* mutant.

Transcripts that were differentially regulated in the UK1 *codY* and 630 *codY* mutants include factors that are directly and indirectly regulated by CodY. To discern which genes may be directly controlled by CodY to influence sporulation, we sought to define

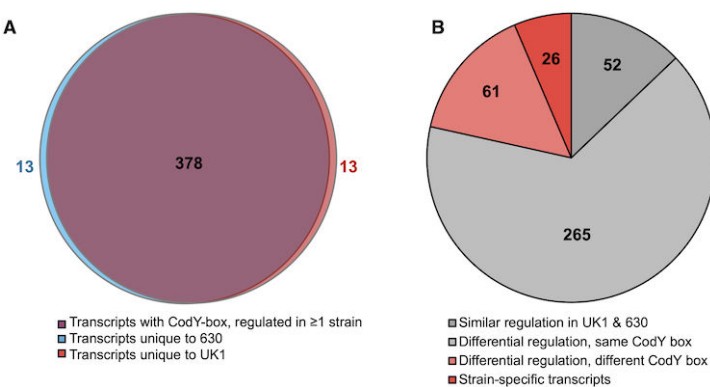

**FIG 2** Representation of CodY-regulated transcripts with putative CodY boxes in strains UK1 and 630. (A) Venn diagram illustration of CodY-regulated genes and operons from RNA-seq data of strains UK1 and 630 that contain predicted CodY-box regulatory elements. A total of 404 transcripts that were differentially regulated ≥3-fold in UK1 Δ*codY*, 630Δ*erm* Δ*codY,* relative to their parent strains, are represented. (B) Representation of CodY-regulated transcripts from A categorized by regulation characteristics.

genes with CodY-binding motifs (CodY boxes). Using the CodY binding sites previously identified in *C. difficile* (13, 14, 37) and potential CodY boxes identified based on the classical gram-positive CodY consensus (AATTTTCWGAAAATT) (38, 39), we narrowed the list of differentially regulated genes to those most likely to be directly regulated by CodY. The resulting list included 404 transcripts with prospective CodY-binding sites within the promoter or coding sequence that were threefold differentially expressed in at least one of the *codY* mutant strains relative to the parental control (Fig. 2; Table S3).

Of the genes and operons listed in Table S3, 52 were similarly regulated by CodY in the UK1 and 630Δ*erm* strains, which limits their likelihood for strain-specific, CodY-dependent impacts. While some of these factors may differ in protein similarity or function that results in differences in sporulation outcomes, such differences were outside the scope of this study. Of the 352 transcripts in Table S3 that were dissimilarly CodY regulated between UK1 and 630Δ*erm*, 265 had identical CodY boxes, which suggests that the differences in expression observed were not due to variation in the inherent ability of CodY to bind to these target sequences. We focused further on the 87 CodY-regulated transcripts with significant differences in expression between the UK1 and 630Δ*erm* strains (Fig. 2B and Tables 1 and 2). Table 1 includes CodY-regulated genes with associated CodY boxes that differ in expression at least twofold between strains, while Table 2 contains CodY-regulated genes that are unique to the genome of either strain. As expected from the sporulation phenotypes of the *codY* mutants, sporulation-specific transcripts comprised many of the genes differentially expressed in the UK1 *codY* mutant (Table S3 ~10%) (33, 40), many of which were late-stage sporulation or germination factors. Unfortunately, increased late sporulation gene expression in UK1 *codY* is not helpful for understanding how CodY differentially regulates the initiation of sporulation, which is controlled by the activation of the master sporulation regulator, Spo0A (33, 41). One factor that is directly involved in Spo0A activity and demonstrated reduced expression in UK1 *codY* is *spo0E*. Spo0E interacts with Spo0A to limit Spo0A activation, which prevents sporulation initiation in *C. difficile* (42). However, the putative CodY boxes that potentially impact *spo0E* were identical in UK1 and 630, implying that the CodY-dependent effect on *spo0E* transcription in the UK1 *codY* mutant was not due to strain-specificity in CodY binding. In addition, a large proportion (20%, Table S3) of the CodY-regulated transcripts in both strains are genes of unknown function, which limits our understanding of their contribution to CodY-dependent phenotypes.

Though few sporulation initiation-associated genes were identified in these data that would clearly explain the increased Spore formation found in the UK1 *codY* mutant, there were notable differences in the expression of genes indirectly associated with greater

**TABLE 1** Differentially expressed CodY target genes in strains 630Δerm and UK1

| | UK1 | | | | 630Δerm | | | | |
| Genetic region | Predicted CodY box[a] | Predicted CodY target | ΔcodY/WT | Genetic region | Predicted CodY box | Predicted CodY target | ΔcodY/WT | Gene names | Putative function |
| --- | --- | --- | --- | --- | --- | --- | --- | --- | --- |
| CDIF27147_00336-00337 | AATATTCAAATAATT AACTTTAGGAAAAA AT AATTTTTTGAAAAAA | CDIF27147_00336-00337 | 16.5–18.3 | CD02130-02140 | AATATTCAAATAATT AACTTTAAGAAAAAT AATTT TTTGAAAAAA | CD02130-02140 | 1.43–1.77 | | Sporulation |
| CDIF27147_00351-00353 | AATTTTCTGACAAAT | CDIF27147_00352-00353 | 0.23–0.28 | CD02260-02280 | AGTTTTCTGCACAGCT | CD02270-02280 | 1.57–4.41 | fliN | Motility |
| CDIF27147_00374-00397 | AACTTTAGAAAAATA AAGTTTATGAAAA TT AATTTTGAGAAAAAT | CDIF27147_00382-00397 | 0.39–0.95 | CD02450-02630 | AACTTTGGAAGATA AAGTTTATGAAAATT AATTT TGAGAAAAAT | CD02670 | 1.21–4.13 | flg, fli, mot, flh | Motility |
| CDIF27147_00476-00478 | CATTTTAGAAAAATT | CDIF27147_00478 | 1.30–3.28 | CD03350-03370 | CATTTTAAAAAAATT | CD03370 | 0.59–8.06 | | Unknown |
| CDIF27147_00481-00482 | AAATATCTGAAAAAA AATTTACTAAAAA CT AAGTTTATGAAAAT GATTTTATGC AAATT | CDIF27147_00481-00482 | 0.24–0.27 | CD03400-03410 | AATTATCTGAAAAAA AATTTACTAAAAACT AAGTT TATGAAAAAT GATTTTATGCAAAT | CD03400-03410 | 1.43–1.62 | | Unknown |
| CDIF27147_00526-00530 | AAAATTCGGAAAATT | CDIF27147_00526-00530 | 10.05–23.88 | CD04450-04490 | AAAATTCAGAAAATT | CD04450-04490 | 0.93–1.34 | oraSE, orr | Amino acid |
| CDIF27147_00566-00567 | AGATTTGTGAAAATA AATTTTGAAAATA GT AATATTTTAAAAATC ATCTTTCTCAC AATT AAGTTTCAAGAAAATA AACTTACT AAAAATC | CDIF27147_00566 | 1.12–6.02 | CD04830-04840 | AGATTTGTGAAAATA AATTTTGAAAATAGT AATAT TTTAAAAATC ATCTTTCTCACAATT AAGTTTCAA GAAATA AACTCACTAAAAATC | CD04830 | 1.38–1.45 | | Transporter |
| CDIF27147_00618 | AACATTCTGAAAAAT AATATAACGAAAA TT AAGTTAAAGAAAAT | CDIF27147_00618 | 8.64 | CD05500 | AATATTCTGAAAAAT AATATAACGAAAAT AAGTT AAAGAAAATT | CD05500 | 1.08 | | Unknown |
| CDIF27147_00723-00725 | AATATACTTTAAATT AACCTTTATATAAATT AGTTTGAAAAAAATT AATTATATCAAAATC GATTTTATG TATT AATTTACAGATACTG AAAGTT GAATTT AATTTACAGATACTG AAAGTT CAGATATT AGATTTATGAAGATT AAT TTGCAGAACTAT CATTACCTGAAAAAT AAATATGTGAAAAAT AACTTTCAGAGA TTA | CDIF27147_00723-00725 | 2.10–4.13 | CD06490-06510 | AATATACTTTAAATT AACCTTTATAAAATT AGTTTG AAAAAAATT AATTATATCAAAATC GATTTTATGG AATTT AATTTACAGATACTG AAAGTTCAGATATT T AGATTTATGAAGATT AATTTGCAGAACTAT CAT TACCTGAAAAAT AAAATATGTGAAAAAT AACTTTC AGAGATTA | CD06490-06510 | 1.01–1.10 | | Peptidases |
| CDIF27147_00734-00738 | AATGTTGTCAAAATT AATTTGATGAAAAT A AATTTTAAATAAAT AAGTTACAGAA AATA ACTTTTCTCAAAATA AATTCCCA GAAAATA TATTTTCTAAAAATC AATTTT TTCAAAATA GATTTTGTGAAAAAT AAG TTTCTGAAATT | CDIF27147_00734-00738 | 49.0–259.4 | CD06590-06630 | AATGTTGTCAAAATT AATTTGATGAAAATA AATTT TCAAATAAAT AAATTACACAGAAATA AATCCCA GAAAATA TATTTTCTAAAAATC AATTTTTTCAAA ATA GATTTTGTGAAAAAT AAGTTTCTGGAAATT | CD06590-06630 | 5.02–436.3 | tcdRBEA | Toxin |
| CDIF27147_00739 | TATTTTAGGAAAATA | CDIF27147_00739 | 33.2 | CD06640 | TATTTTCCTAAAATA | CD06640 | 5.81 | tcdC | Toxin |
| CDIF27147_00772 | AATTATAAGAAGATT | CDIF27147_00772 | 10.3 | CD06910 | AGTTTATAAGAAGATT | CD06910 | 0.43 | | Metabolism |
| CDIF27147_00939-00943 | AATTTGATGAAATTT AATTTTTAAAAAGT T AATTTACGGCAAATG | CDIF27147_00939-00943 | 0.23–0.53 | CD08530-08560 | AATTTGATGAAATTT AATTTTTAAAAAGTT AATTT ACAGCAAATG | CD08530 | 0.58–1.12 | oppBCAD | Metabolite transporter |
| CDIF27147_00947-00949 | TATTATCTGAAAATA AAGTTTTAGAAAATT T | CDIF27147_00948-00949 | 1.23–2.12 | CD08610-08630 | TATTATCTGAAAATA AAGTTTTAGAAACTT | CD08620-08630 | 0.27–0.79 | | Metabolite transporter |

**TABLE 1** Differentially expressed CodY target genes in strains 630Δerm and UK1 (Continued)

| | UK1 | | | | 630Δerm | | | | |
|---|---|---|---|---|---|---|---|---|---|
| Genetic region | Predicted CodY box[a] | Predicted CodY target | ΔcodY/WT | Genetic region | Predicted CodY box | Predicted CodY target | ΔcodY targetΔcodY/WT | Gene names | Putative function |
| CDIF27147_00969-00973 | AATTTTATGAAAGCT TATTTTTAGAGAAT T AATTTCCTCAAAAGT | CDIF27147_00969-00973 | 4.92-6.74 | CD08820-08860 | AATTTTATGAAAGCT TATTTTTAGAGAATT AATTTC CTCAAAAAT | CD08820-08860 | 1.65-2.48 | glgCDAP | Metabolism |
| CDIF27147_01044-01045 | CTTTTTAGAAAAATT ATTTTTATGAGAAT T AATTTAAGAATATA AAGTTTATTAAA ATT AATATTAGTAAAAATT | CDIF27147_01044-01045 | 3.67-20.0 | CD10280-10290 | CTTTTTTAGAAAAATT ATTTTTATGAGAATT AATTTT AAGAATATA AAGTCTATTAAAATT AATATTAGTAA AGTT | CD10280-10290 | 0.49-1.06 | | Signaling |
| CDIF27147_01075 | AATTATTGAAAAAATT AAATTTCACAAAAT T TATTTCAGGAAAATT | CDIF27147_01075 | 0.23 | CD10540 | AATTATTGAAAAATT AAATTTCACAAAATT TATTTC AGGAAAACT | CD10540 | 0.5 | bcd2 | Metabolism |
| CDIF27147_01249 | | | 0.89 | CD12380 | **AATTTTAGGAACATT** | CD12380 | 4.19 | | Unknown |
| CDIF27147_01280-012820 | AATTTTCAGCATATT AGATTTCTCAAAAT T AATTTTATAAAAAT | CDIF27147_01280-012820 | 0.64-0.80 | CD12660-12680 | AATTTTCAGTATATT **AATTAAAAGAAAATT AATTC TCAGAAAATA** | CD12660-12680 | 2.04-3.21 | | Transporter |
| CDIF27147_01285-01288 | **CAATTCAAAAAATT ATTTTTCTGAAAA** AG AATATCCTGAAAATT AATTTGGAGA AGATT AATTTCCATAAATTT | CDIF27147_01285-01288 | 3.36-8.81 | CD12710-12740 | ATTTTTCTGAAAAAG AATATCCTGAAAATT AATTT GGAGAAGGTT AATTTCCATAAATTT | CD1273-12740 | 0.25-1.10 | topA | DNA processing |
| CDIF27147_01432 | CATTTGAAGAAAAATT AATTTTAAGTATAT T **AATTTTCTTATATTT** | CDIF27147_01432 | 0.29 | CD14120 | CATTTGAAGAAAAATT AATTTTAAGTATATT | CD14120 | 1.16 | | Transcription regulation |
| CDIF27147_01501 | **ACTTTGCAGAAAGTT** | CDIF27147_01501 | 0.28 | CD14750 | **GATATTCAAATAATT** | CD14750 | 0.54 | | Unknown |
| CDIF27147_01665 | | | 49.8 | CD15670 | **AATATTGATAAAATT** | CD15670 | 1.01 | cotG | Sporulation |
| CDIF27147_01721 | ATTTTTCAGACAATT AAATTTTACAAAAAT T AATTTTGCGTAATTT AATTTAACAAA AATT AATTTTATTATAATT | CDIF27147_01721 | 0.30 | CD16160 | ATTTTTCAGACAATT AAATTTTACAAAAT AATTT TGTGTAATTT AATTTAACAAAGATT AATTTTATTA TAATT | CD16160 | 1.29 | | Signaling |
| CDIF27147_01737 | AATTATTGCAAAATT | CDIF27147_01737 | 28.1 | CD16310 | AATTTT**GCA**ATAATT | CD16310 | 0.59 | sodA | Metabolism |
| CDIF27147_01805-01806 | AATTTTCTTTAAATT | CDIF27147_01805-01806 | 0.88-1.67 | CD16940-16950 | **ATTTTTCAAAAACTT AATTTTTCAAAAACT** AATT TTCTTTAAATT | CD16940-16950 | 0.23-0.71 | | Unknown |
| CDIF27147_01855-01856 | AATTATTGGTAAATT | CDIF27147_01855-01856 | 6.03-6.48 | CD17400-17410 | AATTATTGCTAAATT | CD17400-17410 | 0.55 | grdGF | Metabolism |
| CDIF27147_01886 | AATTTTAAAAAAATT **AATTTTTTGAAAA AA C**ATTTTCCTAATATT | CDIF27147_01886 | 0.02 | CD17671-17680 | AATTTTAAAAAAATT CATTTTCCTAATATT | CD17671-17680 | 0.09-0.11 | | Unknown |
| CDIF27147_01913 | AAATTCCTAAAAATT | CDIF27147_01913 | 4.77 | CD17930 | AA**GTG**CCTAAAAATT | CD17930 | 0.52 | | Unknown |
| CDIF27147_01965 | AATTTTACGATATTT ATTTTCGAGAAAAAA T | CDIF27147_01965 | 8.23 | CD18440 | AATTTTACGATATTT **AAAATACAGAAAATT** ATTTT TGAGAAAAAT | CD18440 | 1.02 | | Unknown |
| CDIF27147_02022-02023 | GATTTTCATAACATT **AATTTTCAAAGAT TT AAATTCTAAAAATG** | CDIF27147_02022-02023 | 4.15-6.48 | CD18620-18630 | GATTTTCATAACATT | CD18620-18630 | 0.35-0.58 | | Conjugative Transposon |
| CDIF27147_02031 | AACTTCAGACAAAT | CDIF27147_02031 | 0.11 | CD18710 | AAC**T**CTCAGACAAAT | CD18710 | 0.55 | | Conjugative transposon |
| CDIF27147_02062-02067 | AATTTTTTCTAAATT GATTTGCAGAAAG TT AAGTTTCAGAAGATA | CDIF27147_02062-02067 | 16.8-22.8 | CD19120-19170 | AATTTTTTCTAAATT GATTTACAGAAAGTT AAGTT TCAGAAGATA | CD19120-19170 | 0.18-0.42 | eutABCLME | Metabolism |
| CDIF27147_02068 | AAATTTATAAAAATA | CDIF27147_02068 | 67.5 | CD19180 | AAATTTCTAAAAATA | CD19180 | 0.31 | eutK | Metabolism |

*(Continued on next page)*

**TABLE 1** Differentially expressed CodY target genes in strains 630Δerm and UK1 (Continued)

| | UK1 | | | 630Δerm | | | | | |
|---|---|---|---|---|---|---|---|---|---|
| Genetic region | Predicted CodY box[a] | ΔcodY/WT | Genetic region | Predicted CodY box | Predicted CodY target | ΔcodY/WT | Gene names | Putative function | |
| CDIF27147_02170 | CDIF27147_02170 | **ATATTTACGAAAATT** | 321.8 | CD20000 | | | 0.12 | *ispD* | Metabolism |
| CDIF27147_02368 | CDIF27147_02368 | AATTTTAAGAATATA AAAATTCTGAAAAT T | 33.3 | CD22010 | AATTTT**GAGAATATA AAAATTCTGAAATTT** | CD22010 | 7.11 | | Transporter |
| CDIF27147_02391-02392 | CDIF27147_02391-02392 | ATTATTCAAAAAATT | 2.25-3.91 | CD22310-22330 | TTT**ATTCAAAAAATT** | CD22310-22330 | 0.95-1.35 | *asrABC* | Redox |
| CDIF27147_02414 | CDIF27147_02414 | GAATTACTAAAAATA AAGCTTGTGAAAA GT AATATTCATAAATGT AATTTATTGTA ATTT AATTTTAATAATCTT | 0.64 | CD22520 | GAATTAC**TG**AAAATA AAGCTTGTGAAAAGT AATA TTCATAAATGT AATTTATTGTAATTT AATTTTAATA ATCTT | CD22520 | 5.93 | *kamA* | Metabolism |
| CDIF27147_02424 | CDIF27147_02424 | AATATTCTGAAGATA AAATTACAGATAAA T AATCTTTTGAAAAAG ATTTGACTGAA AAAT AAAATTCAGATAATG | 0.06 | CD22630 | AATA**C**TCGAAGATA AAATTACAGATAAAT AATCT TTTGAAAAAG ATTTGACTGAAAAAT AAAATTCA GATAATG | CD22630 | 1.89 | *prsA* | Metabolism |
| CDIF27147_02479-02480 | CDIF27147_02479-02480 | AATTCTATGAAAATT | 0.63-0.75 | CD23260-23270 | AATTCTAT**A**AAAATT | CD23260-23270 | 3.89-4.50 | *gatAB* | Metabolism |
| CDIF27147_02545-02546 | CDIF27147_02545-02546 | ATTTTTAGAAAGTT AATTTAAAAAAAAA TT | 0.37-0.36 | CD23880-23900 | ATTTT**CTAGAAAGTT AATTTTATGAAGATA** AAATT AAGAAAAAT**A** | CD23880-23890 | 1.18-3.24 | *blaRI* | Antimicrobial resistance |
| CDIF27147_02668 | CDIF27147_02668 | ATTTTTCTGAATATT TATTTTCATATATT AAAATTTCATAAGATT | 2.13 | CD25020 | ATTTTTCTGAATATT TATTTTCATATATT AAATTT**T** ATAAGATT | CD25020 | 6.81 | | Cofactor synthesis |
| CDIF27147_02763 | | AATTTTAAGAAAGTT AATTTTCAATAAG TT | 48.6 | CD25990 | **AATTATATTTAAAATT** | CD25990 | 0.50 | | Transcriptional regulator |
| CDIF27147_02961 | CDIF27147_02961 | AGTATTCTGAAAGTT | 0.32 | CD27870 | AGTATTCTGAAAG**CT** | CD27870 | 0.82 | *cwp84* | Cell surface |
| CDIF27147_02971 | CDIF27147_02971 | AATATTCAGAAAAAA AGTTAGCAGAAG ATT AATTTACTGATAGTA TATTGTCTGA AACTT AATATACACAAAATT | 0.33 | CD27970 | **G**ATATTCAGAAAAAA AA**TT**AGCAGAAGATT AATT TACTGATAATA TATTGTCTGAAACTT AATATAC AC AAAATT | CD27970 | 3.54 | | Cell surface |
| CDIF27147_02995 | CDIF27147_02995 | **AATTTTAAGAAAGT** T AATTTTCAATAAG TT | 10.2 | CD28181 (partial) | AATTTTCAATAAGTT | CD28181 | 1.09 | | Unknown |
| CDIF27147_03022 | CDIF27147_03022 | GATTTTAGGAAAATT AATTCACTGAGAG TT AATTTTCATTTAATT | 78.5 | CD28370 | GATTTTAGGAAAATT AATTTACTGAGAGTT AATTT TCATTTAATT | CD28370 | 5.47 | | Unknown |
| CDIF27147_03138 | CDIF27147_03138 | AATTTGACAAAAATT AAGTTTGAAAAAA AT TATTATCAGAAAGTT | 0.51 | CD30040 | ATTTTGACAAAAATT AAGTTTGAAAAAATT TATTA TCAGAAAGTT | CD30040 | 4.40 | *kdgT2* | Metabolite transport |
| CDIF27147_03140 | CDIF27147_03140 | AATATTCTGTATATG AAATTAGTGAAAAT T | 0.41 | CD30060 | AATATTCTGTATATT AAAATTAGTGAAAATT | CD30060 | 4.60 | | Metabolism |
| CDIF27147_03156-03157 | CDIF27147_03156-03157 | AATGTTCCTAAAAAC ATATTTTAGAAAAT T | 101.4-174.3 | CD30230-30240 | AATGTTCCTAAAAAT ATATTTTAGAAAAT T | CD30230-30240 | 0.21-0.37 | | Unknown |
| CDIF27147_03165 | CDIF27147_03165 | ATTTTTTATAAAATT AATTCTTTGAAAAA T | 37.1 | CD30320 | ATTTTTTATAA**G**AT T AATTCTTTGAAAAAT | CD30320 | 0.93 | | Cofactor synthesis |
| CDIF27147_03235-03236 | CDIF27147_03235-03236 | AATTTATTTAGAATT | 1.50-2.46 | CD30970-30980 | AATTTATTTA**A**AATT | CD30970-30980 | 5.28-6.89 | *bglGF* | Metabolite transport |

*(Continued on next page)*

**TABLE 1** Differentially expressed CodY target genes in strains 630Δerm and UK1 (*Continued*)

| | UK1 | | | | 630Δerm | | | | |
|---|---|---|---|---|---|---|---|---|---|
| Genetic region | Predicted CodY box[a] | Predicted CodY target | ΔcodY/WT | Genetic region | Predicted CodY box | Predicted CodY target | ΔcodY/WT | Gene names | Putative function |
| CDIF27147_03313-03314 | AATTATAAGCAAATT | CDIF27147_03313 | 2.36–9.10 | CD31510-31521 | AATCATAAGCAAATT | CD31510 | 0.61–1.26 | | Prophage transcription regulation |
| CDIF27147_03355 | AATATTTATAAAATT | CDIF27147_03355 | 14.3 | CD31840 | AAAATTTATAAACTT | CD31840 | 0.82 | *dpaL* | Metabolism |
| CDIF27147_03396 | TATTTTCTAATAATT | CDIF27147_03396 | 3.83 | CD32190 | TATTTTTTAATAATT | CD32190 | 0.93 | *hslO* | Stress response |
| CDIF27147_03439-03442 | AAAGTACAGGAAATT | CDIF27147_03439-03442 | 5.71–10.1 | CD32600-32630 | AAAGTACAGGAAATT **AATTGATGGAAATA** | CD32600-32630 | 0.45–1.00 | *pstCAB, phoU* | Transporter, transcription regulation |
| CDIF27147_03542 | GATTTTCTGAAAAGA GAATTTCAAAAA AGT | CDIF27147_03542 | 0.26 | CD33690 | GATTTTCTGAAAAAA AAATTTCAAAAAAGT | CD33690 | 5.63 | | Unknown |

[a]Differences in CodY boxes are noted in bold.

**TABLE 2** Unique direct CodY-regulated factors present in the 630Δ*erm* or UK1 strains

| Genetic region | Predicted CodY box | Predicted CodY target | Δ*codY*/WT | Gene names | Putative function |
|---|---|---|---|---|---|
| 630Δ*erm* | | | | | |
| CD02110-02120 | AATTTGATGAAAATA GATTTTCGGAAAAAT | CD02110-02120 | 2.08–3.89 | *licC* | Metabolism |
| CD02410-02440 | ATTTTTTTTAAAATT TATATTCTAAAAATT GATTTTCTGATAATG | CD02410-02440 | 4.31–8.55 | | Motility |
| CD03790-CD03810 | AATATACGGAACATT | CD03790-03810 | 0.34–0.60 | | Conjugative transposon |
| CD04090-04120 | AAATTTCATAAAAAT | CD04090-04120 | 0.33–1.11 | | Conjugative transposon |
| CD04230 | AATTTTCAAAGACTT AAATTACAGAAAAAT AAATTTCTAAAAATG AATATGCTGAAAATC | CD04230 | 0.32 | | DNA replication |
| CD04352 | TACTTTCAGAACATT | CD04352 | 0.27 | | Conjugative transposon |
| CD10921-10940 | ACTTTACAGAAGATT | CD10921-10940 | 0.23–0.27 | | Conjugative transposon, transcriptional regulator |
| CD11030 | AAGTGTCAGAAAATG | CD11030 | 0.32 | | Conjugative transposon |
| CD18510-18550 | GACTTTCTCAAAATT | CD18510-18550 | 0.30–0.68 | | Conjugative transposon |
| CD18840 | AATTTTTATAATATT | CD18840 | 0.07 | | Unknown |
| CD18860 | AATTTTAGGATTATT AATTTACAGCAACTT | CD18860 | 4.47 | | Transcription regulator |
| CD26170 | AATATTCCAAAATTT | CD26170 | 5.33 | | Unknown |
| CD31360-31380 | AATTTTATGATGATT ATTTTTATGAAAATT AATTTACTAAAGATT | CD31360-31380 | 2.95–5.86 | *bglA7F5G4* | Metabolism |
| UK1 | | | | | |
| CDIF27147_00347-00350 | ATTTTCCTGAAAAAT | CDIF27147_00350 | 0.23–0.59 | *rfbBCAD* | Metabolism |
| CDIF27147_ 00657-00658 | AATTTTCTTAATATT | CDIF27147_00657-00658 | 9.18 | | Signaling |
| CDIF27147_00757 | ACTTAACTGAAAATT | CDIF27147_00757 | 24.0 | | Amino acid metabolism |
| CDIF27147_ 01970-01972 | AACTTTTGGAAAAGT | CDIF27147_01972 | 1.49–7.30 | | Conjugative transposon |
| CDIF27147_ 02077-02078 | AATTTACTAAAAATA AATATTGAGAAAAAT | CDIF27147_02077-02078 | 1.08–3.09 | | Metabolism |
| CDIF27147_03267 | AATATTCAGGAACTT | CDIF27147_03267 | 1.56–3.34 | | Metabolism |
| CDIF27147_ 03305-03309 | AATTTTTAAAATATT GATTTTATGAAAATA AATGTTAGGAAAATT AATTTATGGAAGATT ACTTTTAGGAAAATA AGTTTTTAGAAACTT ATATTTTAGAAAATT | CDIF27147_03305-03309 | 0.29–0.50 | | CRISPR |
| CDIF27147_03444-03445 | AATTTTCTCATAATC | CDIF27147_03444-03445 | 4.54–5.27 | | Transporter |
| CDIF27147_03612 | AATTTTCAAAAGAT AATTTGGAGAAGATT | CDIF27147_03612 | 0.29 | | Unknown |
| CDIF27147_03617 | AATTTTCTGATGATG | CDIF27147_03617 | 4.00 | | Unknown |
| CDIF27147_03628 | AATTTTTTAAAACTT AATTTTTACAAAAAT | CDIF27147_03628 | 7.05 | | Unknown |
| CDIF27147_03629 | AATTTGCAAAAGATT AATTTTTATAAACTT | CDIF27147_03629 | 10.8 | | Transposase |
| CDIF27147_03815-03818 | CATTTTTGGAAACTT | CDIF27147_03818 | 2.31–6.48 | | Transposon |

sporulation. The transcriptional analyses revealed significant changes in CodY regulation between strains, including increased relative expression of dozens of metabolic genes in UK1 *codY* that are not observed in 630 *codY*. Furthermore, several of the metabolism loci that are upregulated in UK1 *codY* contain differences in their putative CodY boxes compared to 630 *codY* (Table 1), while some are only encoded by one strain (Table 2). The extensive differences in UK1 *codY* and 630 *codY* metabolic gene expression suggest that these strains have altered responses to nutrient limitation, which may affect the ability to initiate or complete spore formation.

## Repression of multiple direct CodY-regulated factors impacts sporulation in strain UK1

Given the limited information available on the function of many CodY-regulated factors, we selected an assortment of genes present in both strains that were greatly induced or repressed by CodY for further investigation of their impacts on sporulation (Table 3). To determine which directly regulated CodY-dependent transcripts may impact spore formation, we employed a CRISPR interference (CRISPRi) approach to suppress transcription of target genes (43). The UK1 and UK1 *codY* strains were used for these experiments due to the robust CodY-regulated sporulation phenotype in this background. The UK1 strain was used to evaluate the effects of repressing eight CodY-induced factors, while the UK1 *codY* mutant was used to examine repression of six CodY-repressed factors. Strains were transformed with plasmids containing each CRISPRi sgRNA target expressed from a nisin-inducible promoter and grown on 70:30 agar with 1 µg/mL nisin to assess the impact of transcript repression on sporulation (44, 45). The repression of target genes was examined by qRT-PCR during active growth, which confirmed that the targeted transcripts were reduced in all the strains tested (Fig. S1). The sporulation frequencies of strains carrying each sgRNA target were determined after 24 h, as previously noted, and normalized to the respective parent carrying the vector control (pKD). As shown in Fig. 3, suppression of two of the eight CodY-induced transcripts in strain UK1 resulted in significant increases in sporulation relative to the control. The repression of *CDIF27147_01510*, a gene of unknown function, resulted in a ~40-fold increase in sporulation in strain UK1. The expression of *CDIF27147_01510* was reduced approximately 50-fold in the UK1 *codY* mutant and 20-fold in the 630 *codY* mutant (annotated *CD630_14850* in 630) under sporulation conditions (Table S3). The *CD630_14850* gene is controlled by the iron-responsive regulator, Fur, and induced by cysteine, suggesting it is involved in metabolism (46, 47). Similarly, knockdown of the *CDIF27147_02672* transcript led to ~35-fold greater sporulation in UK1 (Fig. 3). Expression of *CDIF27147_02672* was decreased fourfold in the UK1 *codY* mutant and approximately threefold in the 630 *codY* mutant during sporulation (Table S3). *CDIF27147_02672* is part of a dicistronic operon encoding a pH-dependent transcriptional regulator and transporter we recently characterized (*smrRT; CD630_25050-25060*) that contributes to macrolide and lincosamide resistance (48). SmrR represses expression of the *smrT* transporter, which reduces sporulation and toxin production (48). Expression of SmrRT and CDIF27147_01510 does not appear to directly link to Spo0A activity based on known interactions (42) but more likely support cellular homeostasis through pH or nutritional adaptations, respectively.

The UK1 *codY* mutant was used to assess repression of six CodY-repressed factors by CRISPRi and examine their effects on sporulation, as outlined above. Of the six genes assessed in UK1 *codY*, suppression of *CDIF27147_02081* and *CDIF27147_02803* dramatically reduced spore formation (Fig. 4). Repression of *CDIF27147_02081* led to a ~150-fold decrease in sporulation, while knockdown of *CDIF27147_02803* resulted in ~35-fold lower spore formation than the control. *CDIF27147_02081* and *CDIF27147_02803* both encode predicted membrane proteins of unknown function that are expressed during sporulation (33, 49–51). *CDIF27147_02081* expression increased 248-fold in the UK1 *codY* mutant during sporulation, but was down 14-fold in the 630 *codY* mutant (*CD630_19280*) (Table S3). Similarly, *CDIF27147_02803* expression increased 47-fold in UK1 *codY* and decreased threefold in 630 *codY* (*CD630_26360*) during sporulation. To determine if

the impacts of these transcripts on sporulation are specific to the *codY* mutant, we expressed both knockdown constructs in the wild-type UK1 strain and assessed their effects on sporulation (Fig. S3). We found that repression of either *CDIF27147_02081* or *CDIF27147_02803* led to comparable reduction in sporulation as observed in the *codY* mutant, suggesting that both transcripts are important for spore formation. These results and the contrasting expression profiles for these genes in the UK1 *codY* and 630 *codY* mutants suggest that both factors support robust spore formation, but further investigation is needed to understand their roles in sporulation.

## DISCUSSION

While 630 and UK1 encode identical CodY proteins that can bind to the same target sites, the activity of CodY in these backgrounds may be influenced by many factors that cannot be easily measured. CodY regulation is contingent on the availability of the cofactors GTP and BCAA, which trigger conformational changes in CodY that are necessary for DNA binding (11, 13, 14, 27–29). The availability of GTP and BCAA signals amino acid and energy levels in the cell, which can vary in strains based on their ability to take up nutrients or their capacity to utilize nutrient sources. The UK1 and other 027 isolates grow more poorly than the 630 strain in complete defined minimal media (CDMM), and 027 ribotype isolates demonstrate a narrower metabolic repertoire than 630 and many other strains (52–57). The metabolic range of the 027 isolates relative to other strains may contribute to differences in CodY activity. For example, if BCAA are available to bind CodY, even if other growth-limiting nutrients are unavailable, CodY-DNA binding could persist, restricting adaptation to nutrient limitation and decreasing spore formation (Fig. 1, UK1 24 h). Thus, deletion of *codY* in UK1 could expand metabolite availability through nutrient gene derepression to support sporulation. Our data suggest that at least some of the CodY-regulated genes in UK1 repress sporulation, as indicated

**TABLE 3** CodY-regulated genes of UK1 selected for knockdown

| Direct CodY-induced targets | | | | |
|---|---|---|---|---|
| **Predicted CodY target** | **Predicted CodY Box** | **ΔcodY/WT** | **Name** | **Putative function** |
| CDIF27147_01886 | AATTTTAAAAAAATT AATTTTTTGA AAAAA CATTTTCCTAATATT | 0.02 | | Unknown |
| CDIF27147_01510 | CATTATCAGAAAAAT | 0.022 | | Unknown |
| CDIF27147_02271 | AGTTTTTGAAAAATT | 0.04–0.04 | | Transcription regulation |
| CDIF27147_02499 | AAATATCAAAAACTT | 0.12 | | Transcription regulator |
| CDIF27147_00584 | AATATGCAGAAAATG AATTTTCTAT AAATA AAAGTTCTGAAAATA AAT TATGTGAAAATA | 0.18 | | Transcription antiterminator |
| CDIF27147_00748 | ATATTTCATAAAATT | 0.19 | *blaI* | Transcription regulator |
| CDIF27147_03455 | AACTTAATGAAAACT AATATTGACA AAATA AATATCCAGAAATAT | 0.22–0.24 | *spo0E* | Sporulation initiation |
| CDIF27147_02672 | ATTTTTCAAAAATTT | 0.24–0.30 | *smrR* | Transcription regulator |
| Direct CodY-repressed targets | | | | |
| CDIF27147_02081 | AATCTTCAAAAAATA | 248.6–376.2 | | Unknown |
| CDIF27147_00252 | AATCTTAATAAACTT | 267.7 | | Unknown |
| CDIF27147_01772 | AAATTTATGAATATT | 65.9 | | Unknown |
| CDIF27147_02803 | GATTTTTAGAAGATT | 47.4 | | Unknown |
| CDIF27147_01821 | AAATCTCAGAAAGTT | 42.3 | | Metabolism |
| CDIF27147_03734 | ATTCTTATGAAAATA AATGTTAATAA AGTT AATATTTAGAATAAT | 41.4 | | Unknown |

by the hypersporulation of the UK1 *codY* mutant, while in the 630Δ*erm* strain, only the timing of sporulation is advanced in the absence of *codY* (Fig. 1). Overall, the evidence suggests that nutrient availability differs in these strains, leading to differential CodY regulation of sporulation and metabolic processes.

Our data show that the CodY regulons of the 630Δ*erm* and UK1 strains are considerably different (Tables S1 through S3). Additionally, we identified several CodY-dependent genes with putative CodY boxes that differ in these strains (Table 1) and unique CodY-regulated factors present only in one strain (Table 2). Though we were able to identify several factors that are differentially regulated by CodY that have potential CodY-binding sites, further investigation is needed to determine if CodY is the major regulator of these factors and if CodY binds to these boxes. It is also important to note that by limiting our analysis to factors that were differentially expressed in the *codY* mutants by more than threefold, we may have missed some direct CodY-regulated factors that impact sporulation.

Our work demonstrates that multiple factors regulated by CodY can influence sporulation, as illustrated by the phenotypes observed from repression of CodY-regulated factors (Fig. 3 and 4). As CodY regulates hundreds of genes, innumerable effects of global changes in gene expression in the absence of *codY* may contribute to the different sporulation phenotypes in the UK1 and 630Δ*erm* strains. The effects of CodY on sporulation may be an indirect result of altering the nutrients available or cellular functions that are necessary for adapting to post-exponential growth. Many of the CodY-dependent factors that are differentially regulated have no identified function in *C. difficile*, and their roles in sporulation are not known. Further characterization of these CodY-regulated factors, especially those that affect sporulation when repressed, could provide targets for preventing spore formation.

## MATERIALS AND METHODS

### Bacterial strains and growth conditions

*C. difficile* strains were cultivated in a Coy anaerobic chamber at 37°C with an atmosphere of 10% $H_2$, 5% $CO_2$, and 85% $N_2$ as previously described (58). *C. difficile* strains grew in BHIS broth with addition of 0.1% of taurocholic acid (TA, Sigma-Aldrich) to induce

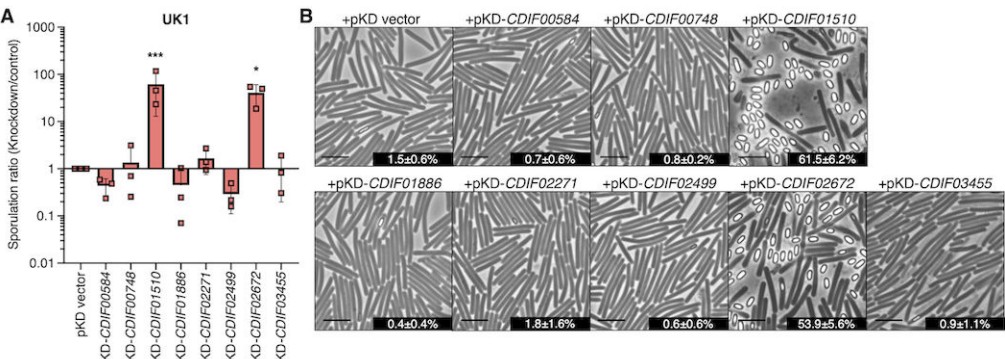

**FIG 3** Repression of specific direct CodY-induced factors increases sporulation in UK1. (A) Ratio of ethanol-resistant spore formation of strain UK1 expressing CRISPRi knockdown constructs relative to a vector control. UK1 carrying pKD-*CDIF01886* (MC2187), pKD-*CDIF01510* (MC2188), pKD-*CDIF02271* (MC2189), pKD-*CDIF02499* (MC2190), pKD-*CDIF00584* (MC2191), pKD-*CDIF00748* (MC2192), pKD-*CDIF03455* (MC2194), pKD-*CDIF02672* (MC2263), and the pKD vector (MC2186) were assessed for spore formation after 24 h growth on sporulation agar (70:30 with 2 µg/mL thiamphenicol, 1 µg/mL nisin). The means, individual values, and standard deviations of ratios (knockdown/control) for at least three biological replicates are shown. (B) Phase-contrast micrographs of the strains in A with sporulation frequencies. Scale bar = 5 µm. The mean, standard deviations, and SEM are shown for three biological replicates. Data were analyzed using a one-way ANOVA, followed by Fisher's LSD. *$P < 0.05$ and *** $P < 0.001$.

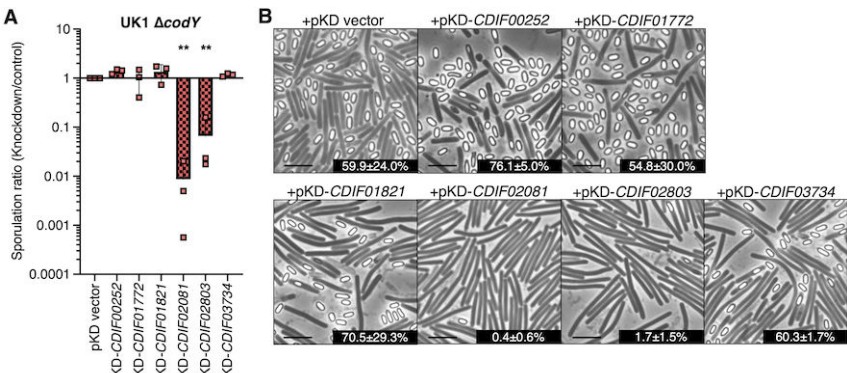

**FIG 4** Repression of specific direct CodY-repressed factors reduces sporulation in the UK1 codY mutant. (A) Ratio of ethanol-resistant spore formation of strain UK1 Δ*codY* mutant expressing CRISPRi knockdown constructs relative to a vector control. UK1 Δ*codY* carrying pKD-vector (MC2195), pKD-*CDIF00252* (MC2196), pKD-*CDIF01772* (MC2197), pKD-*CDIF01821* (MC2219), pKD-*CDIF02081* (MC2216), pKD-*CDIF02803* (MC2218), and pKD-*CDIF03734* (MC2220) were assessed for spore formation after 24 h growth on sporulation agar (70:30 with 2 µg/mL thiamphenicol, 1 µg/mL nisin). The means, individual values, and standard deviations of ratios (knockdown/control) for at least three biological replicates are shown. (B) Phase-contrast micrographs of the strains in A with sporulation frequencies. Scale bar = 5 µm. The mean, standard deviations, and SEM are shown for three biological replicates. Data were analyzed using a one-way ANOVA, followed by Fisher's LSD. **$P < 0.01$.

germination and 0.2% of fructose (D-fructose, Fisher Chemical) to prevent sporulation (59) To maintain plasmids in *C. difficile* strains, 2–10 µg/mL of thiamphenicol was added to cultures. For CRISPRi induction, 1 µg/mL of nisin was added, as needed. *Escherichia coli* strains were cultivated aerobically at 37°C in LB medium (Lennox) with 20 µg/mL of chloramphenicol and/or 100 µg/mL ampicillin (Sigma-Aldrich) for plasmid maintenance. Agar was added at 1.5% for all solid media. *E. coli* was counter-selected post-conjugation with 100 µg/mL of kanamycin.

## Strain and plasmid construction

All plasmids and strains are listed in Table 4. The *C. difficile* strain R20291 027 ribotype genome (GenBank accession no. CP_029423.1) was used as a template for primer construction, and UK1 genomic DNA was used for PCR amplification. To generate sgRNAs, the Benchling CRISPR Guide RNA Design tool was used. sgRNAs were amplified by PCR and cloned into pMC1123 (43, 60). Design details of vector constructions are provided in the supplemental material (Fig. S2).

## Sporulation assays

Sporulation assays were carried out as previously described (66, 67). In short, *C. difficile* cultures at mid-exponential phase (OD$_{600}$ ~0.5) were plated on 70:30 agar supplemented with 2 µg/mL of thiamphenicol and 1 µg/mL of nisin as needed. After 6 (H$_6$), 12 (H$_{12}$), and 24 h (H$_{24}$) of growth, ethanol-resistant sporulation assays were performed as previously described (67). Sporulation frequencies were calculated by dividing the number of spores by the total quantity of cells (spores + vegetative). A *spo0A* mutant was used as a negative sporulation control. For statistical analysis, GraphPad Prism v10.4.1 was used as stated in the figure legends.

## Phase-contrast microscopy

Phase-contrast microscopy was performed at H$_6$, H$_{12}$, and H$_{24}$ as specified in the figure legends using cells grown on 70:30 sporulation agar, as previously described (60).

**TABLE 4** Bacterial strains and plasmids

| Plasmid or strain | Relevant genotype or features | Source, construction, or reference |
|---|---|---|
| Strains | | |
| *E. coli* | | |
| DH5α max efficiency | F− Φ80*lac*ZΔM15 Δ(*lac*ZYA-*arg*F) U169 *rec*A1 *end*A1 *hsd*R17 (rk−, mk+) *pho*A *sup*E44 λ−thi−1 *gyr*A96 *rel*A1 | Invitrogen |
| HB101 | F⁻ *mcrB mrr hsdS20*(r$_B^-$ m$_B^-$) *recA13 leuB6 ara-14 proA2 lacY1 galK2 xyl-5 mtl-1 rpsL20* | Dupuy |
| *C. difficile* | | |
| 630Δ*erm* | Erm$^S$ derivative of strain 630, ribotype 012 | Minton (61) |
| UK1 | Epidemic isolate, ribotype 027 | (62) |
| LB-CD16 | UK1 *codY::ermB* | (63) |
| MC310 | 630Δ*erm spo0A::ermB* | (36) |
| MC364 | 630Δ*erm codY::ermB* | (13) |
| MC855 | 630Δ*erm spo0A::ermB* pMC123 | (41) |
| MC2186 | UK1 pMC1123 | (48) |
| MC2187 | UK1 pMC1170 | This study |
| MC2188 | UK1 pMC1171 | This study |
| MC2189 | UK1 pMC1172 | This study |
| MC2190 | UK1 pMC1173 | This study |
| MC2191 | UK1 pMC1174 | This study |
| MC2192 | UK1 pMC1175 | This study |
| MC2194 | UK1 pMC1177 | This study |
| MC2195 | UK1 *codY::ermB* pMC1123 | This study |
| MC2196 | UK1 *codY::ermB* pMC1158 | This study |
| MC2197 | UK1 *codY::ermB* pMC1160 | This study |
| MC2216 | UK1 *codY::ermB* pMC1156 | This study |
| MC2218 | UK1 *codY::ermB* pMC1162 | This study |
| MC2219 | UK1 *codY::ermB* pMC1163 | This study |
| MC2220 | UK1 *codY::ermB* pMC1164 | This study |
| MC2263 | UK1 pMC1178 | (48) |
| MC3087 | UK1 pMC1156 | This study |
| MC3088 | UK1 pMC1162 | This study |
| Plasmids | | |
| pRK24 | Tra$^+$, Mob$^+$; *bla, tet* | (64) |
| pIA33 | P$_{xyl}$::*dCas9-opt* P$_{gdh}$::sgRNA-*rfp catP* | (43) |
| pMC123 | *E. coli- C. difficile* shuttle vector, *bla, catP* | (45) |
| pMC404 | pMC123 with *catP* replaced by *aad9* | (65) |
| pMC1123 | P$_{cprA}$::*dCas9-opt* P$_{gdh}$::sgRNA-*neg catP*; (pKD) | (60) |
| pMC1156 | P$_{cprA}$::*dCas9-opt* P$_{gdh}$::sgRNA-*CDIF27147_02081 catP* | This study |
| pMC1158 | P$_{cprA}$::*dCas9-opt* P$_{gdh}$::sgRNA-*CDIF27147_00252 catP* | This study |
| pMC1160 | P$_{cprA}$::*dCas9-opt* P$_{gdh}$::sgRNA-*CDIF27147_01772 catP* | This study |
| pMC1162 | P$_{cprA}$::*dCas9-opt* P$_{gdh}$::sgRNA-*CDIF27147_02803 catP* | This study |
| pMC1163 | P$_{cprA}$::*dCas9-opt* P$_{gdh}$::sgRNA-*CDIF27147_01821 catP* | This study |
| pMC1164 | P$_{cprA}$::*dCas9-opt* P$_{gdh}$::sgRNA-*CDIF27147_03734 catP* | This study |
| pMC1170 | P$_{cprA}$::*dCas9-opt* P$_{gdh}$::sgRNA-*CDIF27147_01886 catP* | This study |
| pMC1171 | P$_{cprA}$::*dCas9-opt* P$_{gdh}$::sgRNA-*CDIF27147_01510 catP* | This study |
| pMC1172 | P$_{cprA}$::*dCas9-opt* P$_{gdh}$::sgRNA-*CDIF27147_02271 catP* | This study |
| pMC1173 | P$_{cprA}$::*dCas9-opt* P$_{gdh}$::sgRNA-*CDIF27147_02499 catP* | This study |
| pMC1174 | P$_{cprA}$::*dCas9-opt* P$_{gdh}$::sgRNA-*CDIF27147_00584 catP* | This study |
| pMC1175 | P$_{cprA}$::*dCas9-opt* P$_{gdh}$::sgRNA-*CDIF27147_00748 catP* | This study |
| pMC1177 | P$_{cprA}$::*dCas9-opt* P$_{gdh}$::sgRNA-*CDIF27147_03455 catP* | This study |
| pMC1178 | P$_{cprA}$::*dCas9-opt* P$_{gdh}$::sgRNA-*CDIF27147_02672 catP* | (48) |

**TABLE 5** Oligonucleotides

| Primer | Sequence (5′→3′) | Use/locus tag/reference |
|---|---|---|
| oMC44 | CTAGCTGCTCCTATGTCTCACATC | Forward primer for *rpoC* qPCR (45) |
| oMC45 | CCAGTCTCTCCTGGATCAACTA | Reverse primer for *rpoC* qPCR (45) |
| oMC2618 | GATTATTATGGCGAACAATGAATTAGAAG | Forward primer for *spo0E* qPCR |
| oMC2619 | AAATATTTCTGGATATTCTATGTATGTATTTATCT | Reverse primer for *spo0E* qPCR |
| oMC2362 | AGTTAAACAGAAAGATAATTGCTGTATGG | Forward primer for *smrR* qPCR (48) |
| oMC2363 | ACTTGTAGCCTTACGTTGTTCTTC | Reverse primer for *smrR* qPCR (48) |
| oMC3088 | TTGCAATAAAGTGTGCTATAATTAAACTGTAAATGGCCA | Forward primer to Gibson assemble CRISPRi sgRNAs into pMC1123 (44, 48) |
| oMC3089 | CCTTTTTCTATTTAAAGTTTTATTAAAACTTATAGGATCCGCGGCCGC | Reverse primer to Gibson assemble CRISPRi sgRNAs into pMC1123 (44, 48) |
| oMC3101 | AATTAAACTGTAAATGGCCA<u>AATAATTCCTCACTATCAAG</u>GTTTTAGAGCTAGAAATAGC | Forward primer for sgRNA-*CDIF27147_02081* amplification |
| oMC3103 | AATTAAACTGTAAATGGCCA<u>GAAGAATTACTAAAACTGAG</u>GTTTTAGAGCTAGAAATAGC | Forward primer for sgRNA-*CDIF27147_00252* amplification |
| oMC3105 | AATTAAACTGTAAATGGCCA<u>AATAGTATATTAAAACATAA</u>GTTTTAGAGCTAGAAATAGC | Forward primer for sgRNA-*CDIF27147_01772* amplification |
| oMC3108 | AATTAAACTGTAAATGGCCA<u>ACAACAGTTTCAAGGTCTTG</u>GTTTTAGAGCTAGAAATAGC | Forward primer for sgRNA-*CDIF27147_02803* amplification |
| oMC3109 | AATTAAACTGTAAATGGCCA<u>TTGACTTGGATAGTACCAAG</u>GTTTTAGAGCTAGAAATAGC | Forward primer for sgRNA-*CDIF27147_01821* amplification |
| oMC3110 | AATTAAACTGTAAATGGCCA<u>ATATTTTTGTAAGGATGCAA</u>GTTTTAGAGCTAGAAATAGC | Forward primer for sgRNA-*CDIF27147_03734* amplification |
| oMC3131 | AATTAAACTGTAAATGGCCA<u>TCTTGAAGGTGGTAAATGG</u>GTTTTAGAGCTAGAAATAGC | Forward primer for sgRNA-*CDIF27147_01886* amplification |
| oMC3132 | AATTAAACTGTAAATGGCCA<u>TGGTGACACAAAACAATCCG</u>GTTTTAGAGCTAGAAATAGC | Forward primer for sgRNA-*CDIF27147_01510* amplification |
| oMC3133 | AATTAAACTGTAAATGGCCA<u>GGTATACAAAAGTTTAAGCA</u>GTTTTAGAGCTAGAAATAGC | Forward primer for sgRNA-*CDIF27147_02271* amplification |
| oMC3134 | AATTAAACTGTAAATGGCCA<u>AAAAACGTACCTAAAACTGT</u>GTTTTAGAGCTAGAAATAGC | Forward primer for sgRNA-*CDIF27147_02499* amplification |
| oMC3135 | AATTAAACTGTAAATGGCCA<u>AAAAACGTACCTAAAACTGT</u>GTTTTAGAGCTAGAAATAGC | Forward primer for sgRNA-*CDIF27147_00584* amplification |
| oMC3136 | AATTAAACTGTAAATGGCCA<u>ATATCTTACTTATTGAAGAG</u>GTTTTAGAGCTAGAAATAGC | Forward primer for sgRNA-*CDIF27147_00748* amplification |
| oMC3138 | AATTAAACTGTAAATGGCCA<u>AAATGAGATTGAAGCAGTTA</u>GTTTTAGAGCTAGAAATAGC | Forward primer for sgRNA-*CDIF27147_03455* amplification |
| oMC3139 | AATTAAACTGTAAATGGCCA<u>ATAAAAAAATTATACGTCGA</u>GTTTTAGAGCTAGAAATAGC | Forward primer for sgRNA-*CDIF27147_02672* amplification (48) |
| oMC3235 | TTTCTTAATTATGGCTATGGCAGTT | Forward primer for *CDIF27147_01886* qPCR |
| oMC3236 | ATAAAGGCTTCATAAATACAGCGAA | Reverse primer for *CDIF27147_01886* qPCR |
| oMC3237 | TTGTGTCACCATAAACTTTCCAATA | Forward primer for *CDIF27147_01510* qPCR |
| oMC3238 | AGAGTGATGTTTTTCCTGATGAAAT | Reverse primer for *CDIF27147_01510* qPCR |
| oMC3239 | AACCATCTAAGTTTGGCATCATTAT | Forward primer for *CDIF27147_02271* qPCR |

(*Continued on next page*)

**TABLE 5** Oligonucleotides (*Continued*)

| Primer | Sequence (5′→3′) | Use/locus tag/reference |
|---|---|---|
| oMC3240 | TTTAAGTGCAGAAGGTTATCAAGTT | Reverse primer for CDIF27147_02271 qPCR |
| oMC3241 | AAATGACTTGGCTTCAACAATATTG | Forward primer for CDIF27147_02499 qPCR |
| oMC3242 | AGTAATTGCACGTTCTAATGGTATT | Reverse primer for CDIF27147_02499 qPCR |
| oMC3243 | CAAAGTACGGCCAATTAAATTTTCT | Forward primer for CDIF27147_00584 qPCR |
| oMC3244 | AATTGCTAACCATTCATCTCTTGAT | Reverse primer for CDIF27147_00584 qPCR |
| oMC3245 | AAAGTTGCCACAATCAGAATTAAAG | Forward primer for CDIF27147_00748 qPCR |
| oMC3246 | CGTTTGTTTCCATTGATATTTTTGC | Reverse primer for CDIF27147_00748 qPCR |
| oMC3249 | AGGTTTGACAAGGCTTTCTAAAATA | Forward primer for CDIF27147_00252 qPCR |
| oMC3250 | TCAACCATATTTCCAGCATTTGATA | Reverse primer for CDIF27147_00252 qPCR |
| oMC3251 | CTGCTGTTAATTCAAAATGGAGTTT | Forward primer for CDIF27147_01772 qPCR |
| oMC3252 | ATCTTATCATTTTTATCCTCTCCATT | Reverse primer for CDIF27147_01772 qPCR |
| oMC3394 | CACAATAGCTAAAATTGTGCAATGA | Forward primer for CDIF27147_02803 qPCR |
| oMC3395 | TGCTTATGTTGAAGAAATAGCATCT | Reverse primer for CDIF27147_02803 qPCR |
| oMC3396 | AATGATTTTATTTGGACTTGGAGGT | Forward primer for CDIF27147_01821 qPCR |
| oMC3397 | AATTGCTATTGCTGTTAGAGAATCA | Reverse primer for CDIF27147_01821 qPCR |
| oMC3398 | AGTTGTACCCTCAAAAATATCCATT | Forward primer for CDIF27147_03734 qPCR |
| oMC3399 | ATTTTGTGTTGGATTTTTGGTTCTT | Reverse primer for CDIF27147_03734 qPCR |
| oMC3488 | ACGTTACTATTATTGATAATCTTCACTTATATG | Forward primer for CDIF27147_02081 qPCR |
| oMC3489 | AGATTATAGTACAATAATATAGAAAATTGACACT | Reverse primer for CDIF27147_02081 qPCR |
| 4084 | AACTTATAGGATCCGCGGCCGCTAGTCAGACATCATGCTGATCTAGA | Reverse primer for sgRNAs with NotI site for cloning into pIA33 (43) |

## RNA sequencing (RNA-seq) analysis

*C. difficile* strains were grown on 70:30 agar for 6 h, and cells were scraped, suspended into 1:1:2 ethanol-acetone-water solution, and stored at −70°C prior to processing. RNA was extracted and treated with DNase I (Ambion), as previously described (14, 36). RNA libraries were prepared and processed by the Microbial Genomics Sequencing Center (MiGS; Pittsburgh, PA), as previously described. RNA-seq reads were mapped to the respective reference genome (630; NC_009089.1 and R20291; CP_029423.1) using Geneious Prime v2022.2.2. Expression levels of transcripts were calculated and compared using DESeq2 (68) RNA-seq raw sequence reads were deposited to the NCBI Sequence Read Archive (SRA) as BioProject PRJNA1263881.

## Identification of CodY boxes

Potential CodY boxes were found in the 630 and R20291 genomes from previously published sites, in addition to *in silico* identification (14, 37). The *C. difficile* strain 630 and R20291 genomes (630, NC_009089.1; R20291, CP_029423.1) were screened for the global CodY AATTTTCWGAAAATT consensus sequence containing up to four mismatches using a combination of FIMO MEME and Benchling software (14, 38).

## Quantitative reverse transcription PCR analysis (qRT-PCR)

*C. difficile* strains were grown on 70:30 agar for 6 h, suspended in 1:1:2 ethanol-acetone-water solution, and stored at −70°C. RNA extraction, treatment with DNase I (Ambion), and cDNA synthesis using random hexamers (Bioline) were performed as previously described (14, 36). qRT-PCR was conducted on a Roche LightCycler 96 instrument from 50 ng of cDNA in technical triplicates using Bioline SensiFast SYBR & Fluorescein Mix, with primers shown in Table 5. Expression was normalized to the internal control transcript, *rpoC*, and analyzed using the $\Delta\Delta C_t$ method for relative quantification (69). GraphPad Prism v10.4.1 was used as mentioned in the figure legends for statistical analysis.

## ACKNOWLEDGMENTS

The authors would like to thank members of the McBride lab for useful feedback and suggestions during the course of this work.

This work was supported by the U.S. National Institutes of Health through research grants AI179158 to M.P.M. and AI156052 and AI116933 to S.M.M.

The content of this manuscript is solely the responsibility of the authors and does not necessarily reflect the official views of the National Institutes of Health.

## AUTHOR AFFILIATIONS

[1]Department of Microbiology and Immunology, Emory University School of Medicine, Emory Antibiotic Resistance Center, Atlanta, Georgia, USA
[2]Center for Integrated Solutions for Infectious Diseases, Broad Institute of MIT and Harvard, Cambridge, Massachusetts, USA

## AUTHOR ORCIDs

Adrianne N. Edwards  http://orcid.org/0000-0002-2503-3787
Shonna M. McBride  http://orcid.org/0000-0001-6221-714X

## FUNDING

| Funder | Grant(s) | Author(s) |
| --- | --- | --- |
| National Institute of Allergy and Infectious Diseases | AI179158 | Marcos P. Monteiro |
| National Institute of Allergy and Infectious Diseases | AI156052 | Shonna M. McBride |
| National Institute of Allergy and Infectious Diseases | AI116933 | Shonna M. McBride |

## AUTHOR CONTRIBUTIONS

Marcos P. Monteiro, Conceptualization, Data curation, Formal analysis, Funding acquisition, Investigation, Methodology, Writing – original draft, Writing – review and editing | Adrianne N. Edwards, Data curation, Formal analysis, Investigation, Validation, Writing – review and editing | Michael A. DiCandia, Formal analysis, Writing – review and editing.

## ADDITIONAL FILES

The following material is available online.

## Supplemental Material

**Supplemental figures (Spectrum01706-25-S0001.pdf).** Figures S1 to S3.
**Table S1 (Spectrum01706-25-S0002.xlsx).** RNA-Seq analysis of 630*erm codY* at H6 on 70:30 sporulation agar.
**Table S2 (Spectrum01706-25-S0003.xlsx).** RNA-Seq analysis of UK1 *codY* at H6 on 70:30 sporulation agar.
**Table S3 (Spectrum01706-25-S0004.xlsx).** Differentially expressed genes with predicted CodY boxes in 630 *codY* and UK1 *codY*.

## Open Peer Review

**PEER REVIEW HISTORY (review-history.pdf).** An accounting of the reviewer comments and feedback.

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
