## [Reviewer comments · Microbiology Spectrum]

Microbiology Spectrum

Deciphering strain differences in CodY regulation of *Clostridioides difficile* sporulation

Marcos Monteiro, Adrienne Edwards, Michael DiCandia, and Shonna McBride

Corresponding Author(s): Shonna McBride, Emory University School of Medicine

Review Timeline:

Submission Date:	June 3, 2025
Editorial Decision:	July 6, 2025
Revision Received:	October 9, 2025
Accepted:	October 20, 2025

Editor: Emily Weinert

Reviewer(s): The reviewers have opted to remain anonymous.

Transaction Report:

DOI: <https://doi.org/10.1128/spectrum.01706-25>

Re: Spectrum01706-25 (**Deciphering strain differences in CodY regulation of *Clostridioides difficile* sporulation**)

Dear Dr. McBride:

Thank you for the privilege of reviewing your work and your patience during the review process. Below you will find instructions from the Spectrum editorial office and the reviewer comments.

Revision Guidelines

Sincerely,
Emily Weinert
Editor
Microbiology Spectrum

Reviewer #1 (Public repository details (Required)):

RNA-Seq datasets deposited in the Short Read Archive, though I could not access them. Permissions need to be checked.

Reviewer #1 (Comments for the Author):

The manuscript "Deciphering strain differences in CodY regulation of *Clostridioides difficile* sporulation" investigates the role of CodY in two common research strain of *C. difficile* - 630delta erm and UK1. These are different ribotypes, and notably UK1 is

known for its high virulence and ability to produce toxins and spores. Understanding how sporulation is controlled has practical applications as the spore is the infectious agent for *C. difficile* infection. Therefore, the study is significant and will be of interest to the microbiology community. The manuscript is generally well-reasoned and well-written, though there are places where the writing could be clarified and tightened up. The most significant findings are the results of CRISPR knockdown of CodY-dependent genes that have an effect on sporulation. However, while the authors show that there is a contribution of these genes to sporulation, at least some of these targets have come up previously in other sporulation studies. The direct binding of CodY to target sites is not experimentally shown, and it would be good to note that direct binding is provisional in the text. Below are some major and minor points that, if addressed, will strengthen the manuscript.

Major points

- For the frequency analysis in Fig 1, I'm not certain a t test is appropriate when comparing frequencies. Clearly there are differences. I recommend checking with a stats expert, but perhaps a chi-squared test is more appropriate?
- Lines 127-185: Venn Diagrams (codY/WT) might be a good way to show overlap vs unique differentially regulated genes in 630 vs UK1 strains, as the goal of the study is to highlight the similarities and differences in CodY-dependent gene expression.
- While the authors identify CodY-regulated genes that contribute to sporulation, they do not actually look into how the regulation varies between the two strains of *C. difficile*, which is the premise of the study. That is, no examination of the binding sites, analysis of CodY binding to these regions, etc, is performed.

Minor points

- Lines 69-71: Looks like there is at least one more current review on CodY - I suggest referencing some of the most up-to-date information on the subject. DOI: 10.1007/s00294-016-0656-5

Lines 71-73: Quick check of pubmed shows additional publications that are relevant here. I recommend adding DOI: 10.1093/nar/gkad512, 10.1074/jbc.M116.754309, DOI: 10.1093/nar/gkw775, DOI: 10.1128/JB.05510-11. Might check for other publications as well.

Line 104: "either" doesn't seem appropriate here. Please rephrase.

- Lines 110-112 and figure 1: While the order of magnitude of sporulation in the codY mutant strains relative to their isogenic parent strains in the manuscript text and figure are aligned with regard to the order of magnitudes, the precise numbers don't match. Please make sure they are in agreement. Are these micrographs representative? Same for Figs 2 and 3 (representative?)
- Lines 117-119: "...requires CodY for its full potential." Please clarify, as CodY is a negative regulator of sporulation. By 24 h, the effect of CodY negative regulation on sporulation is gone. To me, this means that whatever CodY is regulating, the target(s) are fully derepressed in that medium.
- Line 122: in contrast to what? Please clarify.
- Line 144: My understanding is that the consensus sequence was first determined in *Lactococcus lactis*. I would reference that publication here as well. DOI: 10.1074/jbc.M502349200
- Lines 160-163: The data referenced are stated to be in Table S3. It's unclear how this supports the argument. Do you mean that, of the putative direct targets shared between UK1 and 630, the UK1 sporulation targets were more strongly de-repressed? Please clarify.
- Lines 207-222: It needs to be clarified that CD630_14850 and CDIF01510 encode the same gene product.
- Line 234: In my opinion, "juxtaposed" isn't quite the right word here. I recommend "contrasting".
- Lines 282-292: I am assuming agar was used at 1.5% w/v to solidify media, but this is not described. Please clarify. Also, please mention that plasmids and strains are listed in Table 4.
- Line 299: It's a little odd that the vector constructions are provided as a figure when a table might better accommodate this information.
- Line 323: The SRA entry is not found when I look it up online. Please check the permissions on the files.
- Line 328: There is a "T" missing from the CodY consensus sequence palindrome.
- Line 336-338: Please provide a justification for the use of rpoC as an internal control transcript.

Reviewer #2 (Comments for the Author):

The manuscript by Monteiro et al describes differential effects on loss of CodY on sporulation in two different *C. difficile* strains of different ribotypes. The experiments produced robust data showing CodY inhibits sporulation in UK1 but not in 630. Most of the experiments are well controlled and the data support the authors' conclusions. Overall, this is a nice study demonstrating strain differences between 630 and UK1. I have only a couple of suggestions to strengthen the conclusions. I also think this is an interesting observation and make a couple of suggestions for ways to further the study but these are not required for this paper.

Major concerns

1. The authors should include complementation of the *codY* mutants in figure 1. I don't doubt the results but given the known issues involving flagellar phase variability and sporulation they should include complementation.
2. The authors suggest that knockdown of the some CodY repressed genes in UK1 (Fig 3) block sporulation. This is clear however, the authors need to knockdown these genes in WT. As is they cannot determine if knockdown generally blocks sporulation or if the effect is specific to *codY* mutants.
3. The authors state that 027 mutants grow poorly in defined media. They speculate in lines 253-254 that *codY* mutants could have increased growth capacities and thus better sporulation. They should directly test this since growth curves are relatively easy.
4. This is more of a question or suggestion for future work but could the authors use GTP or BCAA binding mutants of CodY to dissect the differences between 630 and UK1?
5. Could the authors use CRISPRi to knockdown *codY* in multiple ribotypes to see if CodY effects in 630 or UK1 are more common?

Reviewer #1 (Public repository details (Required)):

RNA-Seq datasets deposited in the Short Read Archive, though I could not access them. Permissions need to be checked.

Reviewer #1 (Comments for the Author):

The manuscript "Deciphering strain differences in CodY regulation of *Clostridioides difficile* sporulation" investigates the role of CodY in two common research strain of *C. difficile* - 630delta erm and UK1. These are different ribotypes, and notably UK1 is known for its high virulence and ability to produce toxins and spores. Understanding how sporulation is controlled has practical applications as the spore is the infectious agent for *C. difficile* infection. Therefore, the study is significant and will be of interest to the microbiology community. The manuscript is generally well-reasoned and well-written, though there are places where the writing could be clarified and tightened up. The most significant findings are the results of CRISPR knockdown of CodY-dependent genes that have an effect on sporulation. However, while the authors show that there is a contribution of these genes to sporulation, at least some of these targets have come up previously in other sporulation studies. The direct binding of CodY to target sites is not experimentally shown, and it would be good to note that direct binding is provisional in the text. Below are some major and minor points that, if addressed, will strengthen the manuscript.

Major points

- For the frequency analysis in Fig 1, I'm not certain a t test is appropriate when comparing frequencies. Clearly there are differences. I recommend checking with a stats expert, but perhaps a chi-squared test is more appropriate?

A chi-square test is generally advised when assessing categorical data, rather than the means of two different populations that are normally distributed. Because there are only two independent populations compared for sporulation frequencies in Fig 1, we performed an unpaired Student's *t*-test.

- Lines 127-185: Venn Diagrams (codY/WT) might be a good way to show overlap vs unique differentially regulated genes in 630 vs UK1 strains, as the goal of the study is to highlight the similarities and differences in CodY-dependent gene expression.

Thank you for this suggestion. To help illustrate the differences in regulation between the strains, we have added a Venn Diagram and a chart (new Figure 2) depicting the overlap of expression between the two strains and the breakdown of CodY-dependent regulation of transcripts.

- While the authors identify CodY-regulated genes that contribute to sporulation, they do not actually look into how the regulation varies between the two strains of *C. difficile*, which is the premise of the study. That is, no examination of the binding sites, analysis of CodY binding to these regions, etc, is performed.

We agree that the next logical investigation to follow this work would be dissection of the specific CodY binding sites and their contribution to regulation and CodY-dependent phenotypes. However, given the large number of transcripts that impacted sporulation and resource constraints, we could

not pursue a detailed examination of CodY binding for this study.

Minor points

- Lines 69-71: Looks like there is at least one more current review on CodY - I suggest referencing some of the most up-to-date information on the subject. DOI: 10.1007/s00294-016-0656-5
- Lines 71-73: Quick check of pubmed shows additional publications that are relevant here. I recommend adding DOI: 10.1093/nar/gkad512, 10.1074/jbc.M116.754309, DOI: 10.1093/nar/gkw775, DOI: 10.1128/JB.05510-11. Might check for other publications as well.

We thank the reviewer for these suggestions and have added the requested references to this section.

Line 104: "either" doesn't seem appropriate here. Please rephrase.

"Either" removed

- Lines 110-112 and figure 1: While the order of magnitude of sporulation in the *codY* mutant strains relative to their isogenic parent strains in the manuscript text and figure are aligned with regard to the order of magnitudes, the precise numbers don't match. Please make sure they are in agreement. Are these micrographs representative? Same for Figs 2 and 3 (representative?)

Yes, the percentage of phase-bright spores can vary from the percentage of cells that survive ethanol treatment, germinate, and outgrow into colonies in the sporulation assay, but the micrographs were taken from the cells prior to assay and are representative of the populations for these strains.

- Lines 117-119: "...requires CodY for its full potential." Please clarify, as CodY is a negative regulator of sporulation. By 24 h, the effect of CodY negative regulation on sporulation is gone. To me, this means that whatever CodY is regulating, the target(s) are fully derepressed in that medium.

CodY has a repressive effect on sporulation, which is most notable during logarithmic growth, where CodY repression is greatest (for WT). However, CodY is a pleiotropic regulator that has complex effects on overall sporulation over time. In the 630 strain, a *codY* mutant generated more spores at early timepoints, but fewer spores than the parent strain by stationary phase, and similar spores by 24 h (H12, Figure 1, Ln 108-116). To clarify, the sentence has been amended to read "These results suggests that CodY suppresses early initiation of sporulation in 630 and that this strain requires CodY to control the timing of sporulation."

- Line 122: in contrast to what? Please clarify.

This sentence has been changed to read: "Thus, the UK1 strain CodY represses sporulation at all growth stages."

- Line 144: My understanding is that the consensus sequence was first determined in *Lactococcus lactis*. I would reference that publication here as well. DOI: 10.1074/jbc.M502349200

Thank you for the suggestion; we have added the above reference.

- Lines 160-163: The data referenced are stated to be in Table S3. It's unclear how this supports the

argument. Do you mean that, of the putative direct targets shared between UK1 and 630, the UK1 sporulation targets were more strongly de-repressed? Please clarify.

I am not sure that I fully understand the question, but what we were explaining in this section is that the UK1 *codY* strain RNA-seq data indicates that there are more sporulation-specific transcripts differentially expressed in the population than samples from the 630 *codY* strain (relative to the parent strains). These results are consistent with the greater spore formation observed in the UK1 *codY* background.

- Lines 207-222: It needs to be clarified that CD630_14850 and CDIF01510 encode the same gene product.

We have added the clarification “annotated as *CD630_14850* in 630”.

- Line 234: In my opinion, "juxtaposed" isn't quite the right word here. I recommend "contrasting". We have changed to “contrasting”

- Lines 282-292: I am assuming agar was used at 1.5% w/v to solidify media, but this is not described. Please clarify. Also, please mention that plasmids and strains are listed in Table 4.

Added: Agar was added at 1.5% for all solid media.

Added: All plasmids and strains are listed in **Table 4**.

- Line 299: It's a little odd that the vector constructions are provided as a figure when a table might better accommodate this information.

Ok

- Line 323: The SRA entry is not found when I look it up online. Please check the permissions on the files.

Thank you for noticing this; we have lifted the embargo.

- Line 328: There is a "T" missing from the CodY consensus sequence palindrome.

Thank you for noticing this oversight.

- Line 336-338: Please provide a justification for the use of *rpoC* as an internal control transcript. *rpoC* and other RNA polymerase component transcripts have been used extensively as normalization controls for relative quantification of transcript abundance by qPCR. We use *rpoC* as an endogenous control because 1) there is considerable precedent for doing so in multiple species, and 2) we find that under most growth conditions the transcription of *rpoC* is robust and consistent.

Reviewer #2 (Comments for the Author):

The manuscript by Monteiro et al describes differential effects on loss of CodY on sporulation id two

different *C. difficile* strains of different ribotypes. The experiments produced robust data showing CodY inhibits sporulation in UK1 but not in 630. Most of the experiments are well controlled and the data support the authors' conclusions. Overall, this is a nice study demonstrating strain differences between 630 and UK1. I have only a couple of suggestions to strengthen the conclusions. I also think this is an interesting observation and make a couple of suggestions for ways to further the study but these are not required for this paper.

Major concerns

1. The authors should include complementation of the *codY* mutants in figure 1. I don't doubt the results but given the known issues involving flagellar phase variability and sporulation they should include complementation.

Complementation of the *codY* mutants was demonstrated in multiple studies and supported the *codY* mutation as the cause of the phenotypes observed (Daou et al, 2019 PLoS ONE 14(1):e026896, Nawrocki et. al, 2016 J Bacteriol 198:2113-2130). In addition, there are published *codY* mutants in many other Firmicutes that have similar impacts on nutritional regulation and sporulation. Because complementation was proven for these specific mutants, we did not include duplication of those efforts in this work.

2. The authors suggest that knockdown of the some CodY repressed genes in UK1 (Fig 3) block sporulation. This is clear however, the authors need to knockdown these genes in WT. As is they cannot determine if knockdown generally blocks sporulation or if the effect is specific to *codY* mutants.

As suggested by the reviewer, we generated knockdowns in the wild-type UK1 background to investigate how repression of specific transcripts effects sporulation in the presence of CodY. As shown in the new Figure S3, we found that repression of either *CDIF27147_02081* or *CDIF27147_02803* led to comparable reduction in sporulation as observed in the *codY* mutant, suggesting that both transcripts are important for spore formation.

3. The authors state that 027 mutants grow poorly in defined media. They speculate in lines 253-254 that *codY* mutants could have increased growth capacities and thus better sporulation. They should directly test this since growth curves are relatively easy.

We were not suggesting that the 027 *codY* mutant would grow better, but rather that the derepression of nutrient genes may help to support sporulation, which is an energy intensive process. Because sporulation is a terminal differentiation event, we would not expect to see improved growth (*i.e.*, increases in cell numbers) in the UK1 *codY* population. To clarify, we have added to this sentence "..., but may not cause apparent changes in cell growth."

4. This is more of a question or suggestion for future work but could the authors use GTP or BCAA binding mutants of CodY to dissect the differences between 630 and UK1?

The CodY protein from these strains are identical, so we are unsure how such mutants could be employed to dissect strain-specific differences. But, the reviewer brings up an interesting point about the availability of GTP and BCAA in these strains, which could explain differences in their outcomes.

5. Could the authors use CRISPRi to knockdown *codY* in multiple ribotypes to see if *CodY* effects in 630 or UK1 are more common?

We are also curious to know how *CodY* impacts sporulation in other ribotypes, but unfortunately, we have had little success introducing constructs into other ribotypes, which is required to perform those studies.

Re: Spectrum01706-25R1 (**Deciphering strain differences in CodY regulation of *Clostridioides difficile* sporulation**)

Dear Dr. McBride:

Thank you for your patience during the review process. Your manuscript has been accepted, and I am forwarding it to the ASM production staff for publication. Your paper will first be checked to make sure all elements meet the technical requirements. ASM staff will contact you if anything needs to be revised before copyediting and production can begin. Otherwise, you will be notified when your proofs are ready to be viewed.

Sincerely,
Emily Weinert
Editor
Microbiology Spectrum